# Robotic Surgery for Biliary Tract Cancer

**DOI:** 10.3390/cancers14041046

**Published:** 2022-02-18

**Authors:** Lyonell B. Kone, Philip V. Bystrom, Ajay V. Maker

**Affiliations:** 1University of Illinois at Chicago–Metropolitan Group Hospitals, Chicago, IL 60607, USA; lyonell.kone@aah.org (L.B.K.); philip.bystrom@aah.org (P.V.B.); 2Department of Surgery, University of California, San Francisco, CA 94143, USA

**Keywords:** cholangiocarcinoma, gallbladder cancer, robotic surgery, biliary tract cancer, klatskin tumor, minimally invasive surgery

## Abstract

**Simple Summary:**

Complete surgical resection of biliary tract cancer provides the best chance at long-term survival. These surgeries are complex and can be associated with high morbidity and prolonged recovery. Minimally invasive approaches have been shown to improve some of these outcomes in other cancers. However, there is a paucity of data in biliary cancer, and even less is known about the outcomes of surgery utilizing a robotic platform. The present review reports the pooled outcomes of robotic surgery for 259 patients with biliary tract cancer. These outcomes are often equivalent to or improved compared to contemporary data on open surgery. The published data lack prospective or randomized studies. Thus, to evaluate any of the potential benefits of robotic surgery for biliary tract cancer, higher-quality studies are needed.

**Abstract:**

Biliary tract cancer consists of cholangiocarcinoma (CC) and gallbladder cancer (GBC). When resectable, surgery provides the best chance at long-term survival. Unfortunately, surgery for these tumors is associated with long operative times, high morbidities, and prolonged hospital stays. Minimally invasive surgery has been shown to impact selected outcomes, including length of stay, in other diseases, and robotic surgery may offer additional advantages compared to laparoscopic surgery in treating bile duct cancers. This is a systematic review of robotic surgery for biliary tract cancer. Predetermined selection criteria were used to appraise the literature. The PRISMA guidelines were followed. In total, 20 unique articles with a total of 259 patients with biliary tract cancer undergoing robotic surgery met the inclusion criteria. For CC and GBC, respectively, the weighted average operative time was 401 and 277 min, the estimated blood loss was 348 and 260 mL, the conversion rate to open was 7 and 3.5%, the all-cause morbidity was 52 and 9.7%, the major morbidity was 12 and 4.4%, the perioperative mortality was 1.4 and 0%, the length of stay was 15 and 4.8 days, the positive margin rate was 27 and 9%, and the number of lymph nodes retrieved was 4.2 and 8. Robotic surgery for biliary tract cancer appears non-inferior to open surgery when compared to the published contemporary data. However, the current literature on the topic is of low quality, and future prospective/randomized studies are needed.

## 1. Introduction

Biliary tract cancer consists of intra-hepatic, peri-hilar, and distal cholangiocarcinoma (CC), along with gallbladder cancer (GBC). The prognosis remains poor (10% overall 5-year survival), and the best chance at long-term survival is surgical resection. Unfortunately, a large proportion of patients are not surgical candidates (80%) at diagnosis. However, in a select group of patients, median overall survival can reach 51 months after resection [1].

CC at the hilum may be classified by the Bismuth-Corlett classification which describes the extent of biliary duct involvement. Type 1 represents the biliary tumor located proximal to the confluence, Type 2 involves the confluence, Type 3a involves both the confluence and the right hepatic duct, Type 3b involves both the confluence and the left hepatic duct, and Type 4 extends into both the right and left hepatic bile ducts [2]. This classification may dictate the type of surgical resection that will be performed. An extra-hepatic biliary resection with biliary-enteric reconstruction is possible for type 1 and 2. However, type 3 will also require a hepatic resection of the caudate lobe or associated hemi-liver. Resections for type 4 tumors are limited by the future liver remnant and reconstructable ducts. Some tumors may also require vascular resection and reconstruction. Localized intrahepatic cholangiocarcinoma may be resected, and, finally, distal CC may require a pancreaticoduodenectomy.

GBC can be classified as incidental, referring to an intraoperative (diagnosis made on frozen biopsy after the gallbladder has been removed but before the surgery is finished) or on permanent section (after the gallbladder has been removed and the surgery finished), or as de novo, referring to the pre-operative diagnosis of GBC. TNM staging is critically important for prognosis but also for management. Incidental GBC with Tis or T1a tumors (limited to lamina propria), and with negative margins does not require an additional liver resection [3]. Surgical management of higher T-stages and de novo GBC, however, may require radical cholecystectomy, which consists of a cholecystectomy, hepatectomy (Segments 4B/5), portal lymphadenectomy, and negative cystic duct margin.

Recently, Sheetz et al. reported that among 169,404 patients, between 2012 and 2018, the use of robotic surgery surged from 1.8% to 15% [4]. Given these rapid changes, there are concerns regarding the broad and indiscriminate implementation of a new surgical platform with limited data. Whereas numerous comparative studies are ongoing for common surgical procedures, the use of a robotic platform for rare and more technically complex surgeries, including biliary tract cancer, is limited.

Indeed, in order to achieve an R0 resection for biliary tract cancer, a surgeon must have a complete understanding of the biliary anatomy and be ready to perform a biliary enteric anastomosis that carries a potential risk of stricture or leakage, a partial or major hepatectomy with risk of hemorrhage or post-hepatectomy liver failure, a vascular resection and reconstruction with risk of hemorrhage or ischemia, a thorough lymphadenectomy of the porta hepatis with risk of achieving inadequate lymph node sampling, and other associated procedures, such as hepatic ultrasonography and bile duct exploration. These technical aspects of the surgery can be challenging, and require proper training, exposure, retraction, and instrumentation, much of which is difficult to do minimally invasively with robotic surgery. Despite these challenges, the first robotic resection of CC was reported by Giulianotti et al. in 2010 [5], and the first robotic GBC resection was reported by Shen et al. in 2012 [6]. There may be some potential advantages to a minimally invasive approach in selected patients, however, and particularly a robotic platform as it pertains to bilio-enteric anastomoses given 3-D magnification and wristed instrumentation. In the present study, we aim to create the most up-to-date systematic review of the literature on robotic surgery for biliary tract cancer.

## 2. Materials and Methods

### 2.1. Search Strategy

A literature search was conducted by two independent researchers (LK and PB) and included all peer-reviewed manuscripts reporting on robotic interventions on cholangiocarcinoma and gallbladder cancer between the first reported case in March 2010 and December 2021. Exclusion criteria included: non-biliary pathology, no original data, no data on robotic technique, non-English manuscript, duplicate data, and staged procedures, e.g., ALLPS. If more than one study was published on the same cohort, only the study with the largest cohort size was included to avoid overlapping populations. The search and selection process, data extraction, and critical appraisal of the selected studies were conducted independently by LK and PB. In the event of discordant findings, a third reviewer was consulted to achieve consensus (AVM).

The following search terms were used with Pubmed *advanced* search feature: (“cholangiocarcinoma” [MeSH Terms] OR “cholangiocarcinoma” [All Fields] OR “cholangiocarcinomas” [All Fields] OR (“klatskin” [All Fields] OR “klatskin s” [All Fields]) OR (“bile duct neoplasms” [MeSH Terms] OR (“bile” [All Fields] AND “duct” [All Fields] AND “neoplasms” [All Fields]) OR “bile duct neoplasms” [All Fields] OR (“bile” [All Fields] AND “duct” [All Fields] AND “cancer” [All Fields]) OR “bile duct cancer” [All Fields])) AND (“robot” [All Fields] OR “robot s” [All Fields] OR “robotically” [All Fields] OR “robotics” [MeSH Terms] OR “robotics” [All Fields] OR “robotic” [All Fields] OR “robotization” [All Fields] OR “robotized” [All Fields] OR “robots” [All Fields]) and (“gallbladder neoplasms” [MeSH Terms] OR (“gallbladder” [All Fields] AND “neoplasms” [All Fields]) OR “gallbladder neoplasms” [All Fields] OR (“gallbladder” [All Fields] AND “cancer” [All Fields]) OR “gallbladder cancer” [All Fields]) AND (“robot” [All Fields] OR “robot s” [All Fields] OR “robotically” [All Fields] OR “robotics” [MeSH Terms] OR “robotics” [All Fields] OR “robotic” [All Fields] OR “robotization” [All Fields] OR “robotized” [All Fields] OR “robots” [All Fields]).

### 2.2. Data Extraction

The data extraction was pre-determined and included the author, Pubmed ID, study type, year of publication, number of patients, type of tumor (intra-hepatic vs. extra-hepatic for cholangiocarcinoma, Bismuth-Corlett for hilar cholangiocarcinoma, and T-stage for GB cancer), age, gender, body mass index, biliary stent placement, biliary drainage procedure, procedure type, use of ICG, operative time, estimated blood loss, conversion to open, all morbidity, major morbidity (Clavien-Dindo ≥ 3), mortality, length of stay, margin status, and number of lymph nodes retrieved.

### 2.3. Data Management

Data are reported as a mean with standard deviations for continuous variables and frequency with percentage for categorical variables. When a study did not provide these estimates but provided the raw data of individual patients enabling the calculation of mean and standard deviation, the results were annotated with a superscript letter “C”. When a study provided median with interquartile range, these estimates were converted to mean with standard deviation using the method by Hozo et al. [7].

The weighted mean was calculated by:$$\mathrm{Weighted mean}\;=\sum_{i}^{n}\left(Xi*Wi\right)/\sum_{i}^{n}Wi$$
with *X* representing the data values to be averaged, *W* the weights applied to each *X* value, and n the number of terms to be averaged.

A meta-analysis was performed with the use of the Cochrane Review Manager 5.4.1. A random effect model was selected. The mean differences with the associated 95% confidence interval were computed. The results were reported in a forest plot.

### 2.4. Critical Appraisal

Independent critical appraisal was performed by two reviewers (LK and PB) using the Joanna Briggs Institute critical appraisal checklist for case reports and case series. After the resolution of any discrepancies between the two reviewers, a traffic light plot was created using the Robvis tool [8]. For the four comparative studies included in this study, we performed a Newcastle-Ottawa Scale critical appraisal. This systematic review was registered with the unique identifying number: “researchregistry7559”.This study was performed in accordance with the Preferred Reporting Items for Systematic Reviews (PRISMA) guidelines.

## 3. Results

The independent review of 146 studies identified 20 unique datasets meeting the pre-determined inclusion criteria (Figure 1). The reviewers had a 95% concordance in their selection process. A third reviewer assisted with resolving any discrepancies. All reviewers approved of the final set of included studies. In total, 11 studies reported data on 146 patients undergoing robotic surgery for cholangiocarcinoma (CC) [5,9,10,11,12,13,14,15,16,17,18], and 10 studies reported data on 113 patients undergoing robotic surgery for gallbladder cancer (GBC) [6,15,19,20,21,22,23,24,25,26].

Among the 146 patients with cholangiocarcinoma, 17 patients had intra-hepatic CC, while the remainder were hilar or unspecified CC, which were further classified by the Bismuth-Corlette classification with 24 (20%) type-1, 11 (9%) type-2, 39 (32%) type-3, 5 (4%) type-4, and 43 (35%) unspecified. The most common procedure was a biliary resection with lymphadenectomy and hepaticojejunostomy (36%); followed by hepatectomy with biliary resection, lymphadenectomy, and biliary reconstruction (29%); hepatic resection without biliary intervention (16%); biliary resection with alternate reconstruction or diversion (15%); and unspecified (4%). Only one study reported the intraoperative use of indocyanine green [9]. One study reported the use of intraoperative cholangioscopy to facilitate extent of biliary resection [10]. The weighted average age of this cohort was 62.5 years old. The weighted average operative time was 401 min. The weighted average estimated blood loss was 348 mL, conversion to open was 7.1%, morbidity rate was 52%, and major morbidity rate (Claviden-Dindo ≥ 3) was 12%. There were two perioperative mortalities (1.4%). The weighted average length of stay was 15.5 days, and the rate of positive margin was 27%. Only three studies reported the number of lymph nodes retrieved, with an average of 4.2 lymph nodes (Table 1).

There were two comparative studies for robotic CC. Yang et al. compared 70 robotic liver resections (only 6 of which included CC) to 252 open liver resections. They reported longer operative time with the robotic approach, but shorter length of stay and lower blood loss [12]. Xu et al. is the only study comparing the robotic (*n* = 10) and open (*n* = 32) approach strictly in patients with CC. They reported longer operative time (703 vs. 475 min) and higher morbidity (90 vs. 50%). There were three major morbidities and one mortality from post-hepatectomy liver failure in the robotic group (Table 2). Given these unfavorable outcomes, the group recommended against the robotic approach for hilar CC. However, it is worth noting that all patients in this cohort underwent extensive resection, including major hepatectomy with biliary resection and reconstruction [17].

Among the 113 patients with gallbladder cancer, 13 were Tis or T1, 64 were stage T2 or greater, and 36 were unspecified. The weighted average age was 63 years old. Whereas 70 patients (62%) underwent en-bloc cholecystectomy with central hepatectomy and portal lymphadenectomy, 32 patients (28%) underwent completion central hepatectomy with lymphadenectomy post a prior cholecystectomy. In total, 11 patients had unspecified procedures. Three studies reported the intraoperative use of indocyanine green in 45 patients. The weighted average operative time was 277 min. The weighted average estimated blood loss was 260 mL. The weighted average conversion to open was 3.5%. The weighted average morbidity rate was 9.7%, while the weighted average major morbidity rate was 4.4%. There was no perioperative mortality reported. The weighted average length of stay was 4.8 days. The weighted average rate of margin positivity was 9%. In total, 9 out of 10 studies reported on the number of lymph nodes retrieved, with an average of 8 lymph nodes retrieved (Table 3).

There were three comparative studies for robotic GB cancer. Tschuor et al. compared 20 robotic to 23 open resections and reported a significant decrease in blood loss and length of stay [20]. Byun et al. compared 16 robotic to 34 open resections and reported a significant decrease in length of stay [21]. Goel et al. compared 27 robotic to 70 open resections and reported lower blood loss, hospital stay, and morbidity with a robotic approach [24] (Table 2). A meta-analysis of those three studies revealed a decrease in estimated blood loss with a mean difference of 360 mL (95% CI: −762,38), and a significant decrease in length of stay with a mean difference of 3 days (95% CI: −4.9, −1.0). There was no clear difference in operative time (Figure 2).

The critical appraisal of all included studies was performed with the Joanna Briggs Institute checklist. Most case series did not have clear inclusion criteria, and few reported consecutive inclusions, resulting in an increased risk of selection bias (Figure 3A). With regards to case reports, three out of four studies did not have clear description of potential adverse events (Figure 3B). Furthermore, the quality of the four comparative studies included in Table 3 were also evaluated with the Newcastle-Ottawa Scale, and each scored 7 out of 9. The major limitation of these studies was the lack of adjustment for potential confounders.

## 4. Discussion

The use of robotic surgery is expanding at an impressive rate. Adoption is, in part, attributed to several technical aspects in comparison to traditional open or laparoscopic surgery, such as improved “ergonomics, decreased surgeon fatigue, tremor filtration, seven degrees of motion, 3-dimensional vision,” and overlay of near-infrared fluorescence (NIRF) imaging [27]. Indeed, Wee et al. recently published a systematic review supporting improved ergonomics and decreased physical strain on the surgeon and trainee with robotic surgery compared to both open or laparoscopic approaches [28]. However, any clinical benefits to the patients have not been well established. Indeed, as of January 2019, in all surgical fields, only 18 randomized control trials on robotic surgery had been published [27]. Furthermore, the financial costs and learning curve remain potential limitations for widespread adoption, particularly around highly technical procedures such as complex hepatobiliary surgery.

The present study is a systematic review of robotic surgery for cholangiocarcinoma and gallbladder cancer. Although there have been previous review articles on minimally invasive surgery for these pathologies, this is the first review strictly focused on a robotic approach. Given the known technical complexity, high morbidity, and mortality of biliary tract surgery for cholangiocarcinoma and gallbladder cancer, it is not surprising to find a limited number of studies on the topic reporting a robotic approach.

When evaluating other reported cohorts of open surgery for CC, the present pooled analysis for robotic surgery does not appear inferior. Indeed, reported in a cohort of 440 patients with hilar CC, Nuzzo et al. a morbidity rate of 47%, mortality rate of 10%, and an R0 resection rate of 77% [29]. Furthermore, the present pooled results for robotic surgery may be compared to the established international benchmark results published in 2021 [30]. Among the 708 cases of hilar CC qualifying as benchmark cases, the median operative time was 432 min, estimated blood loss was 852 mL, length of hospitalization was 16 days, rate all complications was 76%, rate of major complications 57%, in-hospital mortality was 3%, R0 rate was 57%, and lymph node retrieval was 4 nodes [30]. Tang et al. performed a meta-analysis of open vs. MIS surgery for hilar CC which included 9 studies (7 laparoscopic, 2 robotic) and revealed increased operative time and cost with MIS, but a decrease in blood loss, pain, and length of stay [31].

Similarly, when evaluating other reported cohorts of open surgery for GBC, the present pooled analysis for robotic surgery also does not appear inferior. Indeed, Cao et al. reported outcomes of 61 radical cholecystectomies for T1b/T2 GBC and reported an operative time of 216 min, estimated blood loss of 256 mL, length of stay of 11 days, and lymph node retrieval of 8. Furthermore, the same group compared their outcomes to another 61 radical cholecystectomies performed laparoscopically and did not identify any significant differences in perioperative outcomes [32]. Navaro et al. reported 43 open radical cholecystectomies for T2b GBC with an average operative time of 211 min, estimated blood loss of 208 mL, length of stay of 12 days, all morbidity of 11%, and 12 retrieved lymph nodes. Interestingly, this same group also compared these outcomes to laparoscopic radical cholecystectomy and reported a significant decrease in operative time, estimated blood loss, and length of stay. However, these benefits were at the cost of a significant decrease in lymph node retrieval [33].

In summary, the pooled perioperative outcomes of robotic surgery for biliary tract cancer appear non-inferior, if not improved, compared to contemporary studies reporting outcomes of open surgery for either CC or GBC. Certainly, the present study has numerous limitations. Given the limited amount of data on this topic, this review is based primarily on low-quality retrospective case reports or case series with high risk of publication bias and patient selection bias. Indeed, given the unconventional role of robotic surgery in biliary tract cancer, it is likely that the few patients selected to proceed with this novel surgical approach had good physiological reserve and anatomical features, which may account for the observed outcomes. Only four studies comparing robotic to open surgery for biliary tract cancer were identified, therefore limiting our meta-analysis to gallbladder cancer. Furthermore, there remains significant heterogeneity in the surgical management of biliary tract cancer with respect to the extent of resection, approach to biliary drainage, and biliary-enteric reconstruction, rendering published data difficult to compare. Finally, a major factor confounding operative outcomes of robotic surgery pertains to the surgeons’ learning curve and prior experience with the technology. This is a factor that is often not reported and not able to be controlled for in this study.

There is no evidence to suggest that robotic surgery for complex hepatobiliary surgery is slowing down. It is therefore imperative to have higher-quality, prospective, comparative, and randomized studies on this topic. Furthermore, research needs to be dedicated to optimizing the learning curve of robotic surgery. Adjunct technologies that may further facilitate the robotic operator also need to be investigated and further developed. Indeed, NIRF imaging with indocyanine green (ICG) enables visualization and characterization of the biliary anatomy may assist with lymphatic clearance around the porta-hepatis, and with resection of the cystic duct margin. Moreover, NIRF imaging with ICG is an example of a tool that may be better applied on a minimally invasive platform than in a traditional open surgery [22,34].

## Figures and Tables

**Figure 1 cancers-14-01046-f001:**
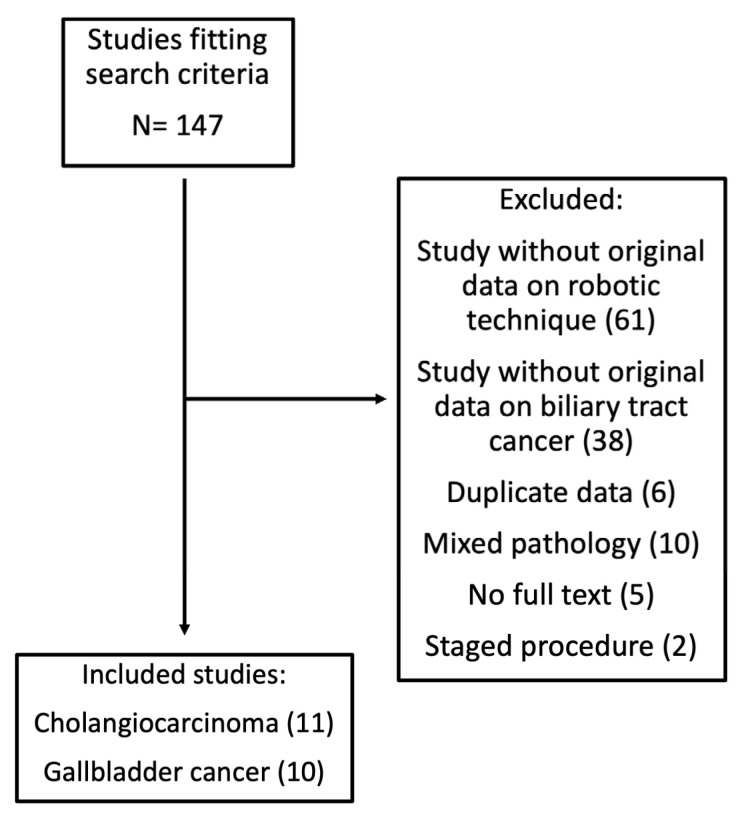
PRISMA flow diagram of study selection process. In total, 20 unique studies were selected, with 1 study meeting the selection criteria for both cholangiocarcinoma and gallbladder cancer.

**Figure 2 cancers-14-01046-f002:**
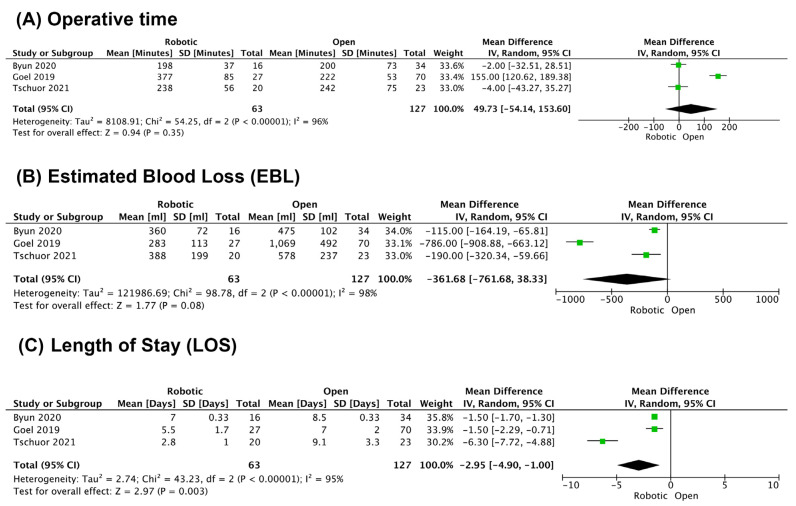
Meta-analysis of robotic vs. open surgery for radical cholecystectomy. (**A**) Operative time in minutes; (**B**) Estimated Blood Loss in ml; (**C**) Length of hospital stay in days.

**Figure 3 cancers-14-01046-f003:**
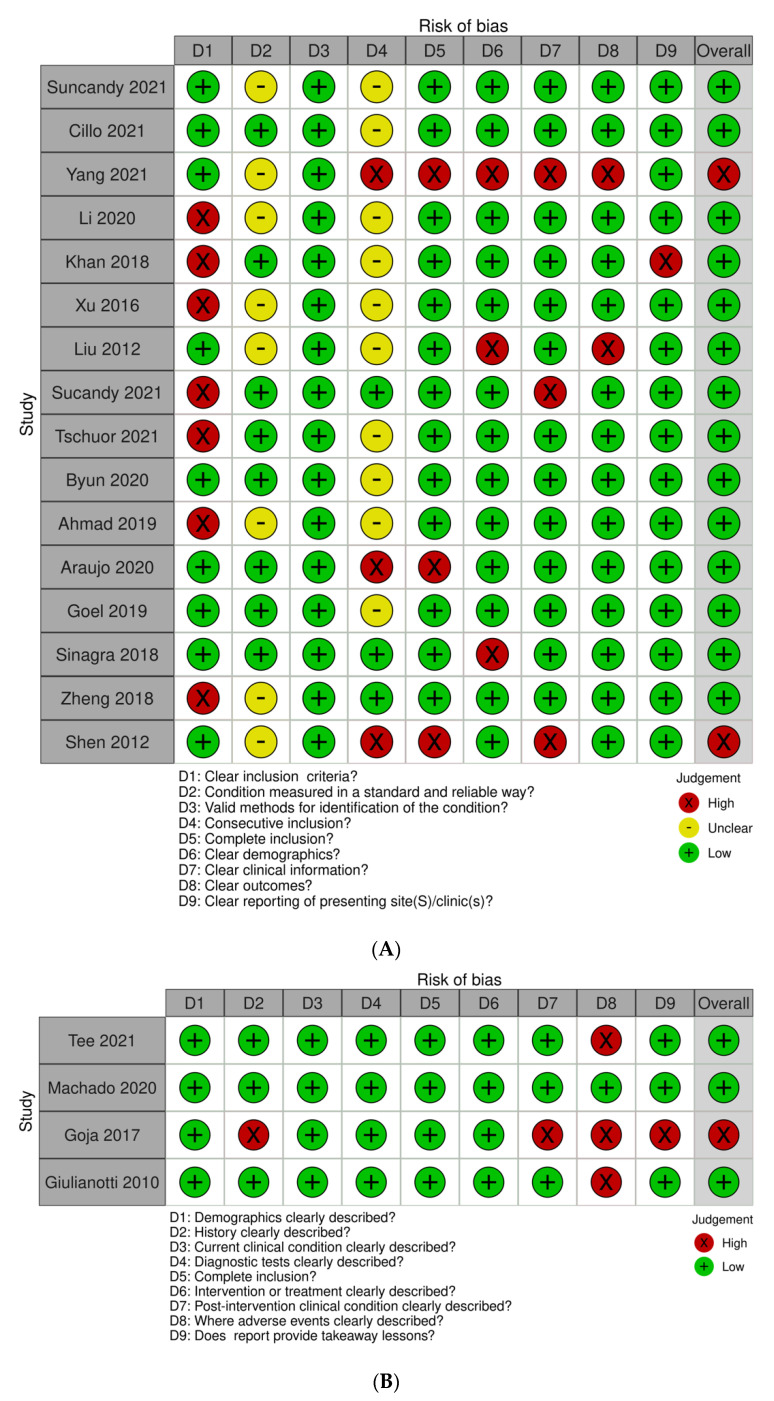
Traffic light plots of Joanna Briggs Institute critical appraisal for case series (**A**) and case reports (**B**) using the Robvis tool which provides copyright permission for publication of created figures [8].

**Table 1 cancers-14-01046-t001:** Systematic review of robotic cholangiocarcinoma.

Author Year	*N*	Study Type	Tumor Type (BC Type)	Age, Male, BMI, BS, BD	Proc., ICG/Cs	Op Time (min)	EBL (mL)	Conv. to Open	All/Major Morbidity	Mort.	LOS	PositiveMargin	LND Retrieval
Tee 2021	1	CR	1 × BC1	58, 1 × M 49, 1 × BS	1 × A 1 × ICG	540	100	0	0/0	0	5	0	12
Sucandy 2021	15	CS	2 × BC1 5 × BC2 6 × BC3 1 × BC4 1 × N/A	74, 9 × M, 24, 12 × BS, 1 × BD	15 × A 15 × Cs	443(85)	182(138)	0	2/1	0	4	4	3.2(2.2)
Cillo 2021	4	CS	4 × BC3	60, 1 × M, 2 × BS, 2 × BD	3 × B 1 × C	850(84) ^c^	700(71) ^c^	1	3/0	0	9(2)^c^	1	-
Yang 2021 ^mp^	6	RC	-	-	-	-	-	1	-	-	-	-	-
Machado 2020	1	CR	1 × BC3	76, 0 × M, 1 × BS	1 × B	480	740	0	1/0	0	-	0	-
Li 2020	48	CS	20 × BC1 6 × BC2 22 × BC3	62, 28 × M, 24, 20 × BD	20 × A ^e^ 28 × B or D ^e^	306(55) ^e^	117(22) ^e^	-	28/5	0	19(8) ^e^	13	-
Khan 2018	16	CS	13 × iCC 3 × hCC	66 ^e^, 5 × M, 28 ^e^	8 × E 8 × F	324(92) ^e^	439(198) ^e^	4	7/3	0	12(6) ^e^	5	4.6(2.2) ^e^
Goja 2017	1	CR	1 × iCC	60, 0 × M,	1 × F	-	-	0	0/0	0	-	0	-
Xu 2016	10	RC	1 × BC1 5 × BC3 4 × BC4	54 ^c^, 8 × M, 6 × BD	6 × B 4 × D	703(62)	1360(809)	0	9/3	1	26(8) ^e^	3	-
Liu 2012 ^mp^	43	CS	3 × iCC 40 × hCC	-	16 × A 6 × F 4 × G 17 × H	-	-	1	-	1	-	-	-
Giulianotti 2010	1	CR	1 × BC3 ^e^	66, 1 × M, 1 × BD & BS	1 × D	540	800	0	0/0	0	11	0	-
Total & pooled Estimates	146		24 × BC1 11 × BC2 39 × BC3 5 × BC4 43 × hCC 17 × iCC	62.5 years old	52 × A 43 × B/D 1 × C 8 × E 15 × F 4 × G 17 × H	All = 401 min hCC = 416 min	All = 348 mL hCC = 330 mL	7/98 (7.1%)	All = 50/97 (52%) Major = 12/97 (12%)	2/140 (1.4%)	All = 15.5 days hCC = 16.3 days	All = 26/97 (27%) hCC = 21/80 (26%)	4.2 LND

BC “Bismuth-Corlette”; BMI “Body mass index”; Proc “Procedure”; ICG “Indocyanine green or reported using Firefly^TM^ system”; Conv. “Conversion”; Mort. “same admission/30-day mortality”; LOS “Length of stay”; LND “lymph node”; CS “case series”; CR “case report”; RC “retrospective cohort”; M “male”; BS “biliary stent”; BD “biliary drainage procedure”; Cs “Intra-operative cholangioscopy”; iCC “intra-hepatic cholangiocarcinoma”; hCC “hilar cholangiocarcinoma”. Superscripts: ^c^ Calculated, ^e^ Estimated, ^mp^ Limited data due to mixed pathology in study. Procedure types: (A) Bile duct resection with hilar lymphadenectomy and hepatico-jejunostomy; (B) Left hepatectomy with bile duct resection, hilar lymphadenectomy, and hepatico-jejunostomy; (C) Left hepatectomy with bile duct resection, hilar lymphadenectomy, and hepatico-gastrostomy; (D) Right hepatectomy with bile duct resection, hilar lymphadenectomy, and hepatico-jejunostomy; (E) Non-anatomic hepatic resection; (F) Major hepatic resection; (G) Bile duct resection with alternate biliary reconstruction; (H) Bile duct resection with biliary diversion.

**Table 2 cancers-14-01046-t002:** Summary of studies that performed comparative analysis.

Cohorts	*N*	Op Time	EBL	LOS	Morbidity	Mortality	R0	LNDR
Xu 2016 (Cholangiocarcinoma)								
Robotic	10	703 min	1360 mL	26 days	9(90%)	1(10%)	N/A	N/A
Open	32	475 min	1014 mL	22 days	16(50%)	2(6%)	N/A	N/A
*p*-value		<0.01	NS	NS	<0.05	NS	N/A	N/A
Tschuor 2021 (Gallbladder cancer)								
Robotic	20	238 min	388 mL	2.8 days	2(10%)	0	16(80%)	6.8
Open	23	242 min	578 mL	9.1 days	4(17%)	0	19(83%)	4.8
*p*-value		NS	0.0019	<0.001	NS	N/A	NS	NS
Byun 2020 (Gallbladder cancer)								
Robotic	16	198 min	360 mL	7 days	1(6%)	0	N/A	7.2
Open	34	200 min	475 mL	8.5 days	5(15%)	1	N/A	7.4
*p*-value		NS	NS	<0.001	NS	NS	N/A	NS
Goel 2019 (Gallbladder cancer)								
Robotic	27	378 min	283 mL	5.5 days	1(4%)	0	27(100%)	11
Open	70	222 min	1069 mL	7 days	15(21%)	0	66(96%)	11
*p*-value		<0.001	<0.001	0.046	0.035	N/A	NS	NS

Op time = operative time, EBL = estimated blood loss, LOS = Length of Stay, R0 = negative surgical margin, LNDR = number of lymph node retrieved. All median estimates were converted to mean.

**Table 3 cancers-14-01046-t003:** Systematic review of robotic gallbladder cancer surgery.

Author. Year	*N*	Study Type	Tumor T-Stage	Age, Male, BMI,	Proc., ICG/Cs	Op Time (SD) min	EBL in mL	Conv. to Open	All/Major Morbidity	Mort.	LOS	Positive Margin	LND Retrieval
Sucandy 2021	15	CS	-	73, 3 × M, 26,	11 × A 4 × B 15 × ICG ^e^	237(86)	222(135)	0	2/0	0	4(4)	2	-
Tschuor 2021	20	RC	2 < T2 18 ≥ T2	64 ^e^, 6 × M,	11 × A ^&^ 9 × B ^&^ 20 × ICG	238(56) ^e^	388(199) ^e^	0	2/2	0	2.8(1) ^e^	4	6.8(2.2) ^e^
Byun 2020	16	RC	5 < T2 11 ≥ T2	64, 10 × M, 25,	16 × A	198(37)	360(72) ^e^	0	1/1	0	7(0.3) ^e^	-	7.2(3.3)
Ahmad 2020	10	CS	10 ≥ T2	69 ^c^, 4 × M,	3 × A 7 × B 10 × ICG	174(45) ^c^	88(65) ^c^	0	1/0	0	3.5 (1.4) ^c^	0	5.4(1.7) ^c^
Araujo 2020	3	CS	3 < T2	45, 1 × M, 31	3 × B	392(16)	186(126)	0	0/0	0	3(0)	0	4.3(1.2) ^c^
Goel 2019	27	RC	5 < T2 22 ≥ T2	54, 9 × M,	25 × A 2 × B	378(85) ^e^	283(113) ^e^	4	1/1	-	5.6 (1.7) ^e^	0	10.8(3.2) ^e^
Khan 2018	11	CS	-	67, 5 × M, 29	-	342(115) ^e^	80(32) ^e^	0	4/1	0	4.8 (1.2) ^e^	2	4.7(1.5) ^e^
Sinagra 2018	3	CS	3 < T2	-	3 × B	283(31) ^c^	200(108) ^c^	0	0/0	0	6(0.8) ^c^	0	21(0.82)
Zeng 2018	3	CS	3 ≥ T2	62, 1 × M	1 × A 2 × B	370(155) ^c^	317(340) ^c^	0	0/0	0	3.3 (0.7) ^c^	0	6.3(5) ^c^
Shen 2012	5	CS	-	57, 2 × M	3 × A 2 × B	200(80) ^c^	210(143) ^c^	0	0/0	0	7.4 (0.5) ^c^	-	8.4(3.4) ^c^
Pooled Estimate	113		13 < T2 64 ≥ T2	63	70 × A 32 × B 45 × ICG	277 min	260 mL	4/113 (3.5%)	All: 11/113 (9.7%) Major: 5/113 (4.4%)	0	4.8 days	8/92 (9%)	8.0 LND

BMI “Body mass index”; Proc “Procedure”; ICG “Indocyanine green or reported using Firefly^TM^ system”; Conv. “Conversion”; Mort. “same admission/30-day mortality”; LOS “Length of stay”; LND “lymph node”; CS “case series”; CR “case report”; RC “retrospective cohort”; M “male”. Superscripts: ^c^ Calculated, ^e^ Estimated, ^&^ five patients did not complete central hepatectomy. Procedure types: (A) En-block cholecystectomy with central hepatectomy and portal lymphadenectomy, (B) Completion central hepatectomy with portal lymphadenectomy (status post prior cholecystectomy).

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
