# Peer review of "Robotic Surgery for Biliary Tract Cancer"

_cancers, 2022, doi:10.3390/cancers14041046_

Round 1

Reviewer 1 Report

Thank you very much for the opportunity to review the manuscript entitled "ROBOTIC SURGERY FOR BILIARY TRACT CANCER". The topic raised describes a complex surgical procedure of hepatobiliar CCC resection by means of roboter assisted methods. The authors review recent literature on this field of interest. 

The statistical analysis and the conclusions of this study is somewhat weak. The odds ratio (ORs) and 95 % confidence interval should be calculated where appropriate. For continuous variables, mean difference is stated, in addition, its 95 % CI would be of interest for the readers.

Although mentioned, the quality of the included studies could have been evaluated by means of the Newcastle-Ottawa Quality scale.

The overall conclusion is, that robotic surgery for biliary tract cancer is not inferior to open surgery - although mentioned, this conclusion lacks discussion of the potential bias of patient selection for the minimally invasive procedure. 

Author Response

  1. The odds ratio (ORs) and 95 % confidence interval should be calculated where appropriate. For continuous variables, mean difference is stated, in addition, its 95 % CI would be of interest for the readers.

We appreciate the opportunity to add to the statistical rigor of the manuscript. The literature search revealed sixteen case report/series that did not perform any form of comparative studies, limiting the ability to perform any form of pooled statistical analysis/ORs as the reviewer suggests. However, in a effort to provide this data, we combined the outcomes data of the four comparative studies available in Table 3. Furthermore, we performed a meta-analysis of the three comparative studies on gallbladder cancer. (See new Figure 2) As suggested by this reviewer, this meta-analysis provides an estimated mean difference with 95% confidence interval for operative time, estimated blood loss, and length of stay. Thank you for this excellent suggestion. 

  1. Although mentioned, the quality of the included studies could have been evaluated by means of the Newcastle-Ottawa Quality scale.

We thank the reviewer for the opportunity to comment on this important point. Indeed, the Newcastle-Ottawa Quality Scale is an excellent measure of study quality that our team has previously used. To our knowledge, this scale is optimally utilized for comparative studies and meta-analyses, therefore, we performed this analysis for the comparative studies that were identified, and this has been added in the critical appraisal section. Given that the majority of the articles utilized in this review were case series and case reviews, and not comparative studies, the Joanna Briggs Institute Critical Appraisal was recommended as the optimal tool to provide analytical evaluations of the quality of the studies.

  1. The overall conclusion is, that robotic surgery for biliary tract cancer is not inferior to open surgery - although mentioned, this conclusion lacks discussion of the potential bias of patient selection for the minimally invasive procedure. 

We thank the reviewer for this comment and agree that discussion of potential bias of patient selection warrants further attention. We have therefore added another statement to this effect in the discussion: “Indeed, given the unconventional role of robotic surgery in biliary tract cancer, it is likely that the few patients selected to proceed with this novel surgical approach had good physiological reserve and anatomical features, which may account for the observed outcomes”

Reviewer 2 Report

The topic is robotic surgery for biliary tract cancer like cholangiocarcinoma or gall bladder cancer. The author collected 259 cases from the literature and described the operation time, morbidity, mortality of the surgeries. Unfortunately, the literature is just the list of the data and the authors did not appropriately compare the data to open surgeries. They just cited the comparison data from the papers. The topic is novel and interesting and the title may be intriguing for many surgeons. However, the manuscript was poorly written. For example, the introduction is just general information about the surgery of biliary tract cancer and does not make readers understand why the authors needed to start the research of robotic surgery.

Author Response

  1. The topic is robotic surgery for biliary tract cancer like cholangiocarcinoma or gall bladder cancer. The author collected 259 cases from the literature and described the operation time, morbidity, mortality of the surgeries. Unfortunately, the literature is just the list of the data and the authors did not appropriately compare the data to open surgeries. They just cited the comparison data from the papers.

We thank the reviewer for this comment and agree that a formal meta-analysis of open or laparoscopic surgery vs. robotic surgery for biliary tract cancer would be a welcomed addition. However, the extensive literature search revealed sixteen case report/series that did not perform any form of comparative studies, limiting an opportunity to perform any form of pooled statistical analysis, therefore, this report is focused on providing a comprehensive descriptive analysis. As it is the first analysis to do so, to the best of the authors’ knowledge, it will provide an iterative step towards more comparative studies as utilization of the technology increases.

Since four comparative studies were identified, at the suggestion of the reviewer, we combined the outcomes data of comparative studies between open and robotic surgery in table 3. Furthermore, we performed a meta-analysis of the comparative studies on gallbladder cancer. Please see new Figure 2.

 Limitations of the available studies has been added in the discussion: “Only four studies comparing robotic to open surgery for biliary tract cancer were identified, therefore limiting our meta-analysis to gallbladder cancer”

  1. the introduction is just general information about the surgery of biliary tract cancer and does not make readers understand why the authors needed to start the research of robotic surgery.

 We thank the reviewer for the opportunity to improve the introduction and agree that additional background is necessary to frame the study. Thus, the introduction now has a greater emphasis on the need to further study/evaluate the role of robotic surgery in biliary tract cancer. The following paragraph was also added: “Recently, Sheetz et al. reported that amongst 169,404 patients, between 2012 and 2018, the use of robotic surgery surged from 1.8% to 15%.[4] Given these rapid changes, there are concerns regarding the broad and indiscriminate implementation of a new surgical platform with limited data. While numerous comparative studies are ongoing for common surgical procedures, the use of a robotic platform for rare and more technically complex surgeries including biliary tract cancer are limited.” This concept is also raised in the discussion.

Reviewer 3 Report

I enjoyed reading your paper and feel that it is timely and helpful, especially for groups that are pushing the envelope. I do however find some of wording confusing so I have a number of suggestions.

Line 18: Change the end of the sentence to read "with long operating times, high morbidities, and prolonged hospital stays".

  • Line 18: Change the sentence to read “surgery for these tumors is associated with long operative times, high morbidities, and prolong hospital stays.”
  • Line 20: “may offer additional advantages”
  • Line 21: “in treating bile duct cancers.”
  • Lines 25-28: Observation: I find it confusing to have the surgical data for cholangiocarcinomas and gallbladder cancers given as CC/GBC, which is the way one would do it if comparing the numerator to the denominator, which is not what is being done. The data is just being given. Thus I would rewrite this section so that one first gives the data for CC and then one gives the corresponding data for GBC. Thus: The weighted average operating time for a cholangiocarcinoma was 401 minutes, the estimated blood loss was 348 mL, the conversation rate to open was 7%, etc.
  • Line 28: I suggest writing “the number of lymph nodes retrieved was …”
  • Line 45: “hepatic duct, Type 4 extends into both the right and left hepatic ducts.”
  • Line 53-54: “GBC can be classified as incidental – referring to an intra-operative (diagnosis made on frozen biopsy after the gallbladder has been removed but before the surgery is done) or on permanent section (after the gallbladder has been removed and the surgery finished) or de novo – referring to pre-operative diagnosis of GBC.”
  • Line 57: “negative margins does not require an addition liver resection.”
  • Line 69: replace “recapitulate” with “do”.
  • Line 74: replace “complete” with “create”.
  • Line 110: replace “lymph node retrieval” with “number of lymph nodes retrieved”.
  • Line 154: replace “lymph node retrieval” with “number of lymph nodes retrieved”.
  • Line 162: Add a space after (B) please.
  • Line 164: Add a space after (E) please.
  • Line 267: Please remove the comma after the period after technology.

Line 20-21: Change the sentence to read "may offer additional advantages compared to laparoscopic surgery in treating bile duct cancers".

Author Response

I enjoyed reading your paper and feel that it is timely and helpful, especially for groups that are pushing the envelope. I do however find some of wording confusing so I have a number of suggestions.

Line 18: Change the end of the sentence to read "with long operating times, high morbidities, and prolonged hospital stays".

  • Line 18: Change the sentence to read “surgery for these tumors is associated with long operative times, high morbidities, and prolong hospital stays.”
  • Line 20: “may offer additional advantages”
  • Line 21: “in treating bile duct cancers.”
  • Lines 25-28: Observation: I find it confusing to have the surgical data for cholangiocarcinomas and gallbladder cancers given as CC/GBC, which is the way one would do it if comparing the numerator to the denominator, which is not what is being done. The data is just being given. Thus I would rewrite this section so that one first gives the data for CC and then one gives the corresponding data for GBC. Thus: The weighted average operating time for a cholangiocarcinoma was 401 minutes, the estimated blood loss was 348 mL, the conversation rate to open was 7%, etc.
  • Line 28: I suggest writing “the number of lymph nodes retrieved was …”
  • Line 45: “hepatic duct, Type 4 extends into both the right and left hepatic ducts.”
  • Line 53-54: “GBC can be classified as incidental – referring to an intra-operative (diagnosis made on frozen biopsy after the gallbladder has been removed but before the surgery is done) or on permanent section (after the gallbladder has been removed and the surgery finished) or de novo – referring to pre-operative diagnosis of GBC.”
  • Line 57: “negative margins does not require an addition liver resection.”
  • Line 69: replace “recapitulate” with “do”.
  • Line 74: replace “complete” with “create”.
  • Line 110: replace “lymph node retrieval” with “number of lymph nodes retrieved”.
  • Line 154: replace “lymph node retrieval” with “number of lymph nodes retrieved”.
  • Line 162: Add a space after (B) please.
  • Line 164: Add a space after (E) please.
  • Line 267: Please remove the comma after the period after technology.

Line 20-21: Change the sentence to read "may offer additional advantages compared to laparoscopic surgery in treating bile duct cancers".

We thank the reviewer for taking the time to provide insightful suggestions for the manuscript, and appreciate the positive comments. The recommended edits have been made as suggested by the reviewer:

  • Line 18 > ok
  • Line 20 > ok
  • Line 21 > ok
  • Line 25-28 > this is a good suggestion. We removed the divider to decrease confusion and explicitly used the “&” sign instead. The word count limit of the abstract limits the ability to report the data of GBC and CC separately.
  • Line 28 > ok
  • Line 45 > ok
  • Line 53-54 > ok
  • Line 57 > ok
  • Line 69 > ok
  • Line 74 > ok
  • Line 110 > ok
  • Line 154 > ok
  • Line 162 > ok
  • Line 164 > ok
  • Line 267 > ok
  • Line 20-21 > ok

Round 2

Reviewer 1 Report

Thank you for the opportunity to read your revised version. I thank the authors for considering my suggestions. I think you have now successfully managed to present the methodological deficits of the individual studies and to present the resulting conclusions of your analysis in a clear way. 

Reviewer 2 Report

The manuscript's topic is robotic surgery for biliary tract cancer like cholangiocarcinoma or gall bladder cancer. The author collected 259 cases from the literature and described the operation time, morbidity, mortality of the surgeries. The revised manuscript is dramatically improved, especially the introduction. Now, the manuscript is eligible to be published. I do not have any suggestions to improve more.